# The Source Matters–Effects of High Phosphate Intake from Eight Different Sources in Dogs

**DOI:** 10.3390/ani11123456

**Published:** 2021-12-04

**Authors:** Britta Dobenecker, Ellen Kienzle, Stephanie Siedler

**Affiliations:** Chair of Animal Nutrition and Dietetics, Department of Veterinary Science, Faculty of Veterinary Medicine, Ludwig-Maximilians-Universität München, Schönleutnerstrasse 8, D-85764 Oberschleissheim, Germany; kienzle@tiph.vetmed.uni-muenchen.de (E.K.); stephanie-siedler@t-online.de (S.S.)

**Keywords:** nutrition, adverse effects, phosphorus, phosphate, PTH, additives, safety

## Abstract

**Simple Summary:**

The vast majority of pet food products on the market contain the major mineral phosphorus in amounts exceeding the recommended daily allowance. A considerable amount of phosphorus in such products is highly water-soluble and bioavailable. Even though a sufficient supply of phosphorus is important for the body, an excessive intake can be harmful, especially in renal patients but also healthy individuals. This study investigated the effects of an excessive intake of organic phosphate sources (e.g., meat and bone meal) and inorganic phosphate salts compared with a balanced control diet without inorganic phosphates on factors of the body’s phosphorus regulatory system in healthy, adult beagle dogs. Most inorganic phosphates but not the organic phosphorus sources caused significant changes in these regulatory factors compared with the control diet. We conclude that the use of these inorganic phosphates in pet food is potentially harmful and should be restricted.

**Abstract:**

Elevated serum phosphate concentrations are an established risk factor for cardiovascular disease and mortality in chronic kidney disease in various species. Independent associations of other parameters of phosphorus metabolism, such as phosphorus intake from different sources and serum concentrations of phosphorus, as well as parameters involved in the regulation, such as parathyroid hormone (PTH) or markers of bone turnover, have been studied in less detail. Therefore, the serum kinetics of phosphate, PTH, and the bone resorption marker bone-specific alkaline phosphatase (BAP) were investigated after 18 days of feeding a control diet and diets supplemented with eight different organic and inorganic phosphate sources aiming at 1.8% phosphorus per dry matter and calcium to phosphorus ratio between 1.3 and 1.7 to 1. Eight healthy beagle dogs (f/m, 2–4 years, 12.9 ± 1.4 kg body weight) were available for the trial. Highly significant differences in the serum kinetics of phosphorus, PTH, and BAP with the highest postprandial levels after feeding highly water-soluble sodium and potassium phosphates were found. We conclude that the use of certain inorganic phosphates in pet food is potentially harmful and should be restricted.

## 1. Introduction

Phosphorus (P) has a vital role in the body. It is involved in numerous metabolic functions and pathways and is stored in the skeleton together with calcium (Ca). For example, its role in energy metabolism via ADP and ATP is crucial to the body’s functionality. Adequate phosphorus balance is a prerequisite for essential cellular functions ranging from energy metabolism to cell signalling. The concentration in blood and tissues is therefore regulated in relatively tight limits, mainly by parathyroid hormone (PTH) and vitamin D3. Several other active substances involved in phosphorus metabolism, such as FGF23 and the Klotho system, are here identified, emphasising the importance of tight regulation.

As seen in advanced renal failure, elevated phosphate levels, together with dysregulated calcium, PTH, and vitamin D levels, contribute to the complex of chronic kidney disease—mineral and bone disease (CKD-MBD). There is even evidence that nutritional phosphate load is a potential risk factor for renal, skeletal, and cardiovascular health in various species, including humans, cats, and dogs [1,2,3,4,5,6,7,8,9,10,11,12]. High serum concentrations of phosphate and linked parameters of phosphorus homeostasis, such as parathyroid hormone (PTH) and fibroblast growth factor 23 (FGF-23), are associated with increased morbidity and mortality in patients with chronic kidney disease [13,14]. Moreover, it is suggested that calcification of vascular cells can occur early in a phosphate-rich environment also in healthy individuals due to a direct causal role of phosphorus in inducing and promoting vascular calcification [15,16]. Moreover, it is described in the literature that a disturbed phosphate balance through excessive intake can inflict irreversible organ damage [7,17,18] and other health problems.

Hyperphosphatemia is a risk factor for developing several different complications in patients with CKD, including secondary hyperparathyroidism and cardiovascular complications, due to the formation of calcium phosphate deposits [19]. Phosphate nephropathy is also known in healthy individuals after inorganic phosphate loading [20,21,22]. Excessive amounts of unbound phosphorus in the blood are excreted via the kidneys. This process is crucial to the body’s healthy function not only because the serum phosphorus concentration correlates to the PTH levels and therefore influences the tightly regulated calcium metabolism via calcitonin and calcitriol, but it also affects bone health. Additionally, phosphorus has a direct impact on soft tissue, such as vascular cells [23], leading to ectopic calcification and cytotoxic effects and, in extreme cases, tissue necrosis [22,24,25]. A high phosphorus excretion via kidneys with an increasing phosphorus concentration in the urine is associated with the detrimental effect of phosphorus excess on renal health [7,10,22,26]. These effects are thought to be due to the harmful action of a high phosphorus concentration on vascular cells and the tubules with calcification and fibrosis [26,27].

Corresponding to water consumption and urine production, phosphate intake determines the concentration of phosphorus in the urine. Various factors, such as moisture content of the diet (i.e., kibble feeding vs. homemade or moist diets), certain diseases, anaesthesia, and species differences in the ability to concentrate the urine, add to this. In this aspect, a cat might have a disadvantage due to the characteristically concentrated urine. A recent study found that in cats, the phosphorus intake preceding the diagnosis of chronic kidney disease (CKD) is significantly higher compared with cats without kidney problems [11]. CKD prevalence is increasing in humans, is already high in dogs and especially cats [28,29,30,31], and is also a leading cause of death [32]. Up to a certain point, the remaining intact glomeruli have a large capacity to mask the receding functionality of the kidneys, and by the time of diagnosis, the proportion of damaged renal tissue is as high as 66–75% [33]. The workload for each nephron increases with the progression of the disease. Therefore, the intake of substances usually excreted in the urine, especially phosphorus, should be limited.

The major regulator in the body for calcium and phosphorus metabolism is the parathyroid hormone (PTH). Keeping calcium and phosphorus in defined boundaries in the blood is critical for the body to provide enough ions for all essential functions, such as muscle contractility, blood coagulation, energy metabolites (ATP), and bone formation, but also to prevent calcification of soft tissues. PTH is secreted by the C-cells of the parathyroid gland. Depending on the situation, the hormone increases the serum calcium concentration by increasing the bone resorption and regulates the renal phosphorus excretion, therefore, narrowing the calcium to phosphorus ratio (Ca/P) in the blood. The sensitivity of PTH receptors is crucial for the efficiency of PTH. Interference of this sensitivity is well known not only in ruminants where high calcium intake antepartum can cause hypocalcemia in early lactating cows [34]. Felsenfeld and Rodriguez [35] were able to demonstrate that in nonruminating species, oral phosphorus intake has a significant impact on the calcemic efficacy of PTH. Feeding high phosphorus diets leads to increased PTH levels in healthy and CRI rats to maintain normal calcium blood levels. This is partly due to the decreased calcium-sensing receptor expression in hyperplastic parathyroid glands [36].

Overall, studies performed in humans and animals have convincingly demonstrated the toxic effects of phosphate in accelerating various pathologies, ranging from vascular calcification to tumour formation and ageing. Diets leading to increased concentrations of phosphorus and PTH in the blood should therefore be avoided.

Dobenecker et al. [37] investigated how two phosphates (NaH_2_PO_4_; KH_2_PO_4_), when added to the diet of healthy adult dogs, affect the parameters of phosphorus homeostasis. While added organic phosphate in the form of poultry meal had no effects when compared with a balanced diet, both highly soluble inorganic phosphates disrupted the calcium and phosphorus homeostasis thus causing potential harm for renal, cardiovascular, and skeletal health.

The aim of this study was to test the effects of increased mid-term phosphorus intake from a wider variety of phosphate sources on the postprandial serum level of calcium, phosphorus, and PTH in dogs compared with the fasted state and to a balanced control diet. Other parameters such as apparent digestibility were evaluated additionally.

## 2. Materials and Methods

### 2.1. Animals

For the study, eight healthy young Beagles (2 m/6 f, 3–5 years of age, 10.3–15.1 (13.3 ± 1.5) kg BW) were involved. The dogs were pair-housed in climate-controlled indoor pens with sufficient resting places with bedding material. Every day of the year, they had access to outdoor runs of about 50 m^2^ equipped with kennels as well as trees or awning, in established groups of 4 to 7 animals for at least 6 h per day (between 8 a.m. and 4 p.m.). Additionally, the dogs were walked on a leash and trained in regular intervals. During the digestibility trials, they were walked at least twice daily. The indoor pens had natural and artificial light for a minimum of 8 h per day, depending on the season. Humidity varied between 40 and ~70%. Fresh air was provided through a ventilation system throughout the year. The indoor temperature was kept above 16 °C.

All dogs had an optimal nutritional status with a body condition score of 5/9. Individual energy requirements were determined over a period of 10 weeks prior to the start of the study by identifying the energy necessary for bodyweight maintenance. The energy was apportioned individually to ensure a constant body weight. Regular weekly weighing with an adaptation of energy supply in case of body weight loss by adding lard ensured minimal weight changes with a constant intake of the basic rations.

### 2.2. Feeding

Three weeks prior to the start of the study, the dogs were fed a complete maintenance food supplemented with casein, gelatin, CaCO_3_, and in some cases, lard to ensure a Ca and P intake according to their individual requirements. After a transition period of 3 d, the dogs were fed a complete and balanced diet (control CON) in an amount that ensured individual energy requirements were met. The diet was produced from thoroughly mixed cooked tripe, cooked rice, and casein (63.8, 31.9, and 4.3%, respectively (Table 1)) and was then individually supplemented with minerals and vitamins to meet the dog’s requirements [38,39]. In this basic diet, the phosphorus intake target was 100 mg phosphorus/kg BW^0.75^ (0.4% phosphorus/DM, 760 mg phosphorus /1000 kcal ME, 180 mg phosphorus /MJ ME) a Ca/P ratio of 1.4/1 without adding phosphorus from other organic or inorganic sources. For 15 days, the dogs received only this diet, and a balancing trial was performed during the last 5 d of this period (14–18 d). On day 18, blood samples were obtained before feeding (≥12 h after the last meal) and 2 h after feeding the full amount of the daily ration. This procedure was repeated, adding different sources of phosphorus and aiming at ingestion of 500 mg phosphorus/kg BW^0.75^ (1.8% phosphorus /DM, 3010 mg phosphorus /1000 kcal ME, 720 mg phosphorus /MJ ME). To allow for eventually higher energy requirements, lard was added to the diet of some dogs. Refusals were weighed and recorded.

Between the trials, there were washout periods of at least 10 d where the control diet CON was fed. As phosphorus sources inorganic phosphate salts (monocalcium phosphate (Ca (H_2_PO_4_)_2_), dicalcium phosphate (CaHPO_4_, synonym: calcium monohydrogen phosphate), monosodium phosphate (NaH_2_PO_4_), sodium tripolyphosphate (Na_5_P_3_O_10_; STTP), monopotassium phosphate (KH_2_PO_4_), potassium pyrophosphate (K_4_P_2_O_7_)) and organic phosphates (poultry meal (PM) and cattle bone meal (CBM)) were used in the following order: CON, diCaP, mNaP, PM, STTP, mCaP, CBM, mKP, KpyrP. Because of the low palatability of sodium tripolyphosphate (STTP; Na_5_P_3_O_10_) and potassium pyrophosphate (K_4_P_2_O_7_) in these cases, the added amount was reduced to about 300 mg phosphorus/kg BW^0.75^ (1.1% phosphorus/DM, 105 mg phosphorus/1000 kcal ME, and 440 mg phosphorus/MJ ME). To exclude a possible influence of particle size of the phosphorus source on phosphorus digestibility, it was ensured through grinding and sieving that all sources had a practically suitable particle size of less than 1mm. To ensure a Ca/P ratio of 1.3–1.4/1, CaCO_3_ was added to the diets with inorganic phosphate salts when necessary. In the diets with organic phosphorus sources such as meat and bone meal, the Ca/P ratio amounted to 1.7/1 without adding CaCO_3_.

### 2.3. Sample Collection and Handling

During the 5 d balancing trial, faeces were collected quantitatively whenever defecation was spotted to reduce the probability of coprophagia, even though this was unlikely in the selected dogs. Urine was sampled about 2 h pre- and 3 to 4 h postprandially using a special scoop. Food and faeces were lyophilised, ground, mixed thoroughly and sampled for analysis. Crude nutrients in food and faeces were determined using the Weende method [40]. For mineral analyses for wet digestion of dry feed samples, faecal samples, and urine samples (frozen until analysis, thawed, and thoroughly stirred) were done in a microwave digestion unit (Ethos 1600, MLS GmbH Leutkirch, Germany).

On the last day of each trial (d18), blood samples were obtained in the fasted animal (>12 h after the last meal) and 2 h postprandially from the Vena cephalica antebrachii and allowed to stand for 20–30 min before centrifugation at 2400 rpm for 10 min. For PTH measurements, 0.5 mL of the serum were transferred to transport tubes with a PTH stabiliser composed of a mixture of several proprietary protease inhibitors of the laboratory (ALOMED, Radolfzell, Germany). Calcium was determined by flame emission spectrometry (EFOX 5053, Eppendorf AG, Hamburg, Germany) and phosphorus photometrically with ammonium molybdate and ammonium vanadate in HNO_3_ (GENESYS 10 UV, Thermo Spectronic, Rochester, NY, USA) [41]. A direct luminometric sandwich immunoassay (ILMA) was performed for the PTH determination, using two polyclonal antibodies against different epitopes of the intact human PTH. The first antibody is focused against the N-terminale epitope and acts through its marking with acridinium ester as a tracer. The second antibody is focused against the C-terminale epitope and is solid bound. The reference range of the laboratory for PTH in adult dogs is 8–45 pg/L. Bone-specific ALP was measured with the human ELISA kit from MicroVueTM Quidel^®^ BAP Enzym-Immunoassay (TECOmedical AG, Sissach, Switzerland) validated for dogs. The reference range for this ELISA kit in adult dogs between 3 to 7 years is 6.7 ± 3.6 U/L [42]. Urine creatinine was determined using the Jaffé method (MicroVue Creatinine Assay Kit, Quidel Corporation, reader: Sunrise Tecan).

### 2.4. Statistics

For statistical evaluation, the first mean values and standard deviation were calculated. Then the software SigmaPlot (Systat Software GmbH, Frankfurt/Main, Germany) was used. In normally distributed (Shapiro–Wilk test) and variance homogenous (Levene test) values, a one-way ANOVA for repeated measures was performed. The remaining values were tested using a nonparametric test (Friedman, RM ANOVA on ranks). All values were then tested against the results from feeding the control diet (Dunn’s method as a post hoc test). Data are given as mean with interquartile range (IQR) in brackets.

## 3. Results

The phosphorus source affected the palatability of the diet when given in this excessive amount. The addition of NaH_2_PO_4_ resulted in the slightest and of KH_2_PO_4_ in the highest reduction of acceptance, followed by STTP, even though the latter was added in reduced amounts. Extension of feeding time, which normally amounted to less than 15 min, was allowed to ensure complete intake, and the reoffering of food led to a feed intake sufficient to prevent relevant weight loss. In all other trials, the dogs consumed the apportioned amount of food in the standard feeding time (~15 min). Overall, the dogs maintained their body weight (<4% variation) and BCS.

The apparent digestibility of phosphorus was 38.7% (23.6/57.4%) in the control diet without added phosphorus. It was neither significantly altered when KH2PO4 (diet mKP) was added, which produced the highest values for apparent phosphorus digestibility, nor after adding CaHPO_4_ (diet mKP), NaH_2_PO_3_ (mNaP), or K_4_P_2_O_7_ (KPyrP) (Table 2). Diets with added organophosphates (PM, CBM) as well as the phosphate salts diCaP (CaHPO_4_) and STTP (Na_5_P_3_O_10_) caused a significantly reduced apparent phosphorus digestibility in comparison to CON (*p* < 0.05). The apparent digestibility of calcium was negative in all diets but diCaP and mKP (*p* < 0.05).

In the serum samples obtained 2 h postprandially, the phosphorus concentrations were significantly higher in all diets where inorganic phosphates were added, compared with diet CON (diCaP and STTP (*p* < 0.01); mNaP, mKP, mCaP, KpyrP (*p* < 0.001) (Table 3). Neither the control diet nor the excess phosphate intake from organic phosphate sources (PM, CBM) caused an increase in postprandial serum phosphorus concentrations. The preprandial serum phosphorus concentrations were within the reference range in all animals but significantly lower in group mNaP and mKP than CON.

Preprandial serum concentrations of PTH were similar and within the reference range in all diets (Table 3). Two hours postprandially, there was a significant increase of PTH values in the groups fed diets with added monophosphates (calcium, sodium, and potassium monophosphates) as well as sodium and potassium polyphosphates (STTP, K_4_P_2_O_7_). The lower phosphorus intake in diet STTP and KpyrP did not prevent this increase in serum hormone concentrations. In all named groups, most of the individuals had PTH values above the reference range. Addition of mNaP caused the highest postprandial serum concentrations, 90 (50/114) pg/mL vs. 20 (19/27) pg/mL in CON.

Feeding a high phosphorus diet led to a significant reduction of preprandial and postprandial serum calcium concentrations in all diets but to a different extent (Table 4). The serum calcium–phosphorus product was affected especially in the postprandial samples (*p* < 0.001) with significantly higher values in groups mCaP, mNaP, mKP and KPyrP (Figure 1).

Because 24 h sampling of urine was not possible in this trial, free catch urine before and after the dogs were fed their daily ration was collected and analyzed. The concentration of phosphorus measured in the urine was related to creatinine in order to quantify the urinary phosphorus excretion. The intake of organophosphates given in amounts sufficient to meet the maintenance requirements (CON) resulted in low pre- and postprandial phosphorus to creatinine ratios (1.0/2.8; Table 5). Especially in the urine samples collected postprandially, significant differences between the feeding groups were detected: in group CON, PM, and CBM, the phosphorus to creatinine ratio remained low whereas all inorganic phosphates led to a significantly higher postprandial ratio, mNaP group with added NaH_2_PO_4_ and mKP with added KH_2_PO_4_ showing the highest ratios.

The bone resorption marker bALP was within the reference range (6.7 ± 3.6 U/L) in all animals but one from group mCaP (preprandial). In the preprandial serum samples, the marker was significantly lower after feeding the control diet and diet PM and STTP than in the other phosphorus excess diets. In the postprandial serum samples, the bALP concentrations were significantly higher in diet diCaP, mNaP, and mKP compared with CON 6.1 (5.1/7.3), 6.5 (5.8/8.4), and 6.5 (4.8/6.2), respectively vs. 3.4 (2.7/4.8) U/L.

## 4. Discussion

The aim of the study was to test which phosphate sources added to a complete and balanced control diet have an impact on parameters of phosphorus homeostasis in healthy dogs. Inorganic salts of phosphoric acid and phosphorus from bones and cartilage, i.e., ‘organic’ phosphates from carcass meal and bone meal, were given in amounts similar to those used in the study of Dobenecker et al. [37]. Total phosphorus amounts of this magnitude can also be found in pet food products on the market [45]. Inorganic phosphates are used in processed feed products for technical purposes, including water binding, preservation, texture, colour, and palatability enhancement. Due to the fact that neither the declaration of the total amount of phosphorus nor the phosphorus sources are mandatory in pet foods, at least in the EU, it is not possible to directly compare the percentages of inorganic phosphate salts and phosphate originating from animal or plant material (‘organic’ phosphates) of the trial diets with products on the market.

The addition of these relatively high amounts of phosphates to a food based on tripe and rice led to reduced palatability in some cases. Therefore, the amounts of NaH_2_PO_4_ and KH_2_PO_4_ added to the control diet had to be reduced. To a certain degree, this compromised the direct comparison with the effects of the other phosphate sources used in this study. Therefore, when comparing the results, it has to be taken into account that the measured effects in diets with added NaH_2_PO_4_ and KH_2_PO_4_ were caused by considerably lower phosphorus intake.

As expected, excessive intake of phosphorus from organic sources led to a significant reduction of the apparent digestibility of phosphorus because of the significant linear relationship of its true digestibility over a wide range of intake in the adult dog [46]. This reduced apparent digestibility of phosphorus in the diets containing organic phosphates can be explained by the fact that phosphorus from these sources is bound to protein, phytate, or other molecules with low solubility, such as mineral complexes. This requires digestion and partial transformation of these phosphate sources before absorption. The consequences on the absorption are also known from human studies showing that organic phosphates are only absorbed with a percentage of 40 to 60%, whereas inorganic phosphates, such as phosphate additives, are 90% to 100% absorbed [47]. Inorganic phosphates, on the other hand, can be absorbed directly, especially when they are soluble. The inorganic phosphates used in this trial can be categorized in uncondensed primary phosphates (mNaP, mCaP, and mKP), secondary phosphate (diCaP), condensed polyphosphate (KPyrP), and linear condensed tripolyphosphate (STTP). The latter must be hydrolyzed before absorption, whereas other phosphates can be absorbed directly [48,49], which is probably what explains the differences in apparent digestibility. There was no systematic effect of the calcium to phosphorus ratio in this trial. The addition of different inorganic phosphate sources led to significant differences in the apparent digestibility of phosphorus, irrespective of the similar calcium to phosphorus ratio of 1.3 and 1.4 to 1 (Table 2). A hypothesis is that phosphate from highly soluble sources, such as mKP, is absorbed before complexes with calcium, which are less digestible, can be formed.

That inorganic phosphates are more readily available than organic ones is reflected in the postprandial serum phosphorus concentrations of the dogs. In the control feeding as well in the groups with a phosphorus excess from organic sources, no postprandial increase in serum phosphate was detected, while the addition of inorganic phosphates led to significantly higher concentrations without exception. The highly soluble mNaP and mKP caused the highest postprandial values, exceeding the reference range of each dog. Hyperphosphatemia has several adverse health effects, such as calcifying soft tissues in kidneys and blood vessels. Phosphate can lower renal and circulating Klotho, an antiaging protein involved in cytoprotection and antifibrosis, with antioxidative and antiapoptotic effects [50,51]. The phosphate-associated higher risk of death, primarily based on information from renal patients, is well documented in various species, including dogs [52,53]. In the groups with the highest serum phosphate concentration, mNaP and mKP, the calcium by phosphorus product exceeded the critical value of 55 mg^2^/dL^2^ [44] in all dogs with 97 ± 15 mg^2^/dL^2^ (mNaP) and 90 ± 11 mg^2^/dL^2^ (mKP) by far. A significantly higher product as in the control was also detected in the postprandial samples of group mCaP and KPyrP (Figure 1). Calcium by phosphorus product of ≥55 mg^2^/dL^2^ increases the risk of soft tissue calcification [44] and is therefore also used in the staging system of the International Renal Interest Society (IRIS) for dogs [54,55]. A product of >70 mg^2^/dL^2^ was made responsible for the calcification of the paws of feline and canine patients with chronic kidney disease [56], and values of >77 mg^2^/dL^2^ are defined as a negative prognostic marker for dogs with CKD [55]. It is not a farfetched conclusion that a continuously and markedly increased serum calcium–phosphorus product of this magnitude might cause adverse health effects also in healthy individuals. Products above the cited thresholds also existed in the other trials with added inorganic phosphate, albeit in fewer cases, while diets CON, PM, and CBM only caused a product between 36 and 39 mg^2^/dL^2^ on average with none of the values above 55 mg^2^/dL^2^.

As expected, the PTH concentration mirrored the phosphate concentrations in the serum of the dogs. PTH is a potent phosphatonin, which counters an increased phosphate burden by accelerated renal P excretion. All inorganic phosphate salts, except diCaP, caused a significantly increased PTH concentration after the meal. That this happened to this extent even in the groups with lower phosphate intake, STTP and K_4_P_2_O_7_, demonstrates that possible consequences of increased PTH concentrations have to be taken into account, especially since the same phosphate excess from organic sources, representing a situation more natural to cats and dogs, did not cause an effect on PTH. It is important to note that a skeletal resistance to PTH and a consequently decreased calcemic response to PTH is linked to dietary phosphorus loading [57].

As a potent phosphatonin, PTH accelerated renal phosphate excretion after feeding several inorganic phosphate salts, as demonstrated convincingly by the difference between pre- and postprandial P/Crea ratios. Compared with the control or the excess phosphate feeding using organic sources, the inorganic phosphate led to a distinct and highly significant increase in postprandial renal phosphate excretion. Because the concentration of phosphorus is linked to adverse effects on kidney tissues with a high P concentration in the urine probably also affecting glomerular endothelial cells [58] and increasing interstitial fibrosis with modification of the kidney autophagy process [59], the phosphate concentration in the urine might need to be limited.

That the bone resorption marker bALP was affected by the amount and source of inorganic phosphate addition to an otherwise balanced diet further underlines the need to adjust the phosphate supply, especially in patients with chronic kidney disease-metabolic bone disorder but also healthy individuals.

The results and implications of this study align well with the findings of Dobenecker et al. [38], where phosphorus kinetics and effects on phosphorus homeostasis was investigated in healthy dogs in a subset of the phosphate sources used in this trial.

## 5. Conclusions

This study demonstrated that in contrast to the more natural organic phosphate sources, excessive supply with the tested inorganic phosphates significantly disrupted elements of phosphorus homeostasis in healthy dogs. Therefore, adverse health effects of ingesting such inorganic phosphates cannot be excluded. Therefore, the use of inorganic phosphate sources, particularly mNaP, mKP, and KPyrP, in dog and cat foods must be considered potentially hazardous.

## Figures and Tables

**Figure 1 animals-11-03456-f001:**
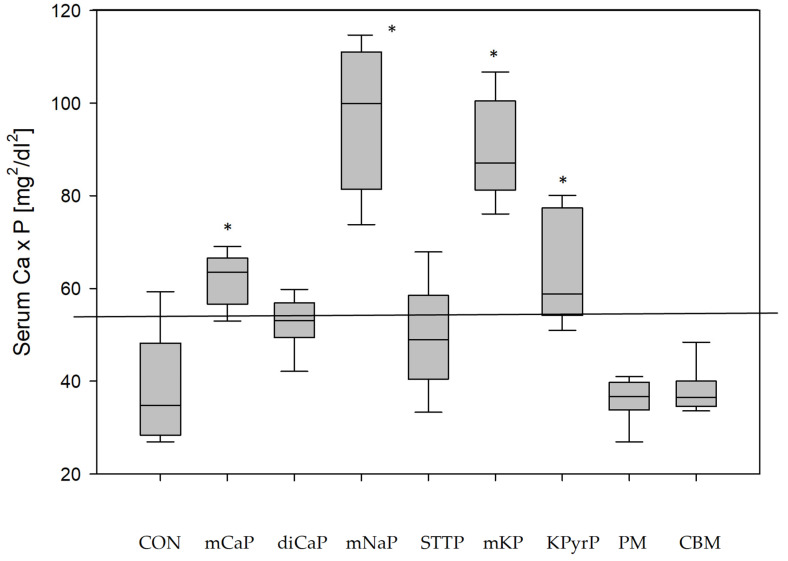
Postprandial serum calcium–phosphorus product (mg^2^/dL^2^) after control and phosphorus excess feeding. * Significantly different from control diet CON (ANOVA *p* < 0.001; vs. CON using Dunn’s Method). Line: 55 mg^2^/dL^2^: threshold after Block et al. [44].

**Table 1 animals-11-03456-t001:** The concentration of phosphorus, calcium, and the Ca/P ratio of the diets.

Diet	Main Phosphorus Source	Phosphorus mg/100g DM	Calcium mg/100g DM	Ca/P
CON	Tripe, rice, casein	443 ± 53	610 ± 72	1.4
mCaP	Ca (H_2_PO_4_)_2_	1826 ± 198	2321 ± 258	1.3
diCaP	CaHPO_4_	1821 ± 119	2381 ± 154	1.3
mNaP	NaH_2_PO_4_	1777 ± 68	2315 ± 82	1.3
STTP	Na_5_P_3_O_10_	1130 ± 151	1529 ± 215	1.4
mKP	KH_2_PO_4_	1871 ± 196	2343 ± 249	1.3
KpyrP	K_4_P_2_O_7_	1154 ± 53	1578 ± 74	1.4
PM	Poultry meal	1963 ± 165	3405 ± 288	1.7
CBM	Cattle bone meal	1696 ± 165	2953 ± 296	1.7

**Table 2 animals-11-03456-t002:** Apparent digestibility of phosphorus and calcium in adult dogs fed diets containing various sources of phosphorus.

	Apparent Digestibility Phosphorus (%)	Apparent Digestibility Calcium (%)
CON	38.7 (23.6/57.4)	−29.9 (−75.4/−0.8)
mCaP	34.6 (32.9/43.0)	2.8 (1.4/17.6) *
diCaP	20.5 (13.8/26.7) *	−5.6 (−20.9/5.0)
mNaP	32.5 (27.5/46.0)	−4.6 (−12.5/4.2)
STTP	18.7 (14.7/31.4) *	−9.0 (−13.7/1.0)
mKP	45.2 (36.8/50.3)	5.0 (−9.8/11.9) *
KpyrP	30.3 (27.2/34.3)	−14.2 (−18.3/−3.7)
PM	8.7 (0.2/13.2) *	−5.3 (−15.9/2.1)
CBM	20.5 (15.5/23.8) *	−2.2 (−3.5/2.8)

Median (IQR); * Significant difference to control group (* *p* < 0.05).

**Table 3 animals-11-03456-t003:** Serum phosphorus (mmol/L) and parathyroid hormone PTH (pg/mL) concentrations in fasted dogs (preprandial) and 2 h after the intake of the daily ration (postprandial).

	Serum Phosphorus (mmol/L)		PTH (pg/mL)	
	Preprandial	Postprandial	Preprandial	Postprandial
CON	1.3 (1.1/1.4)	1.2 (0.8/1.5)	21.5 (19.0/23.5)	20.0 (19.3/27.0)
mCaP	1.2 (0.9/1.3)	2.0 (1.9/2.2) ***	19.4 (18.6/21.6)	42.5 (35.0/61.9) *
diCaP	1.1 (1.0/1.2)	1.6 (1.5/1.8) **	18.0 (16.3/22.5)	27.0 (19.3/35.0)
mNaP	1.0 (1.0/1.1) *	3.1 (2.6/3.6) ***	20.3 (19.4/25.8)	89.8 (49.9/113.8) *
STTP	1.2 (1.0/1.3)	1.7 (1.3/2.1) **	18.4 (16.8/21.0)	43.6 (30.7/74.6) *
mKP	0.9 (0.8/1.0) *	2.8 (2.4/3.1) ***	22.0 (19.6/28.2)	65.4 (48.7/104.4) *
KpyrP	1.2 (1.2/1.3)	2.1 (1.7/2.5) ***	16.2 (13.4/21.2)	64.6 (47.7/100.6) *
PM	1.1 (0.9/1.4)	1.2 (1.0/1.3)	22.6 (18.7/38.1)	26.9 (24.6/30.7)
CBM	1.2 (1.0/1.2)	1.2 (1.1/1.3)	19.9 (17.1/22.8)	23.6 (18.8/31.9)

Reference ranges (specific for the applied method): Phosphorus 0.7–1.6 mmol/L; PTH 8–45 pg/mL. Median (IQR); * Significant difference to control group (* *p* < 0.05, ** *p* ≤ 0.01 and *** *p* ≤ 0.001).

**Table 4 animals-11-03456-t004:** Serum calcium concentrations in fasted dogs (preprandial) and 2 h after the intake of the daily ration (postprandial).

Serum Calcium [mmol/L]		
	Preprandial	Postprandial
CON	2.6 (2.5/2.8)	2.7 (2.6/2.8)
mCaP	2.6 (2.5/2.6) *	2.5 (2.4/2.5) ***
diCaP	2.6 (2.5/2.7)	2.6 (2.5/2.6) *
mNaP	2.6 (2.5/2.6)	2.5 (2.4/2.6) ***
PM	2.6 (2.5/2.6) *	2.5 (2.4/2.6) ***
STTP	2.6 (2.5/2.7)	2.4 (2.4/2.5) ***
CBM	2.6 (2.5/2.7)	2.6 (2.5/2.6) *
mKP	2.6 (2.5/2.6) *	2.6 (2.6/2.7)
KpyrP	2.5 (2.5/2.6) *	2.5 (2.4/2.5) ***

Reference range 2.3–3.0 mmol/L [43]; Median (IQR); * Significant difference to control group (* *p* < 0.05 and *** *p* ≤ 0.001).

**Table 5 animals-11-03456-t005:** Pre- and postprandial urinary phosphorus to creatinine ratio (P/crea) after control and phosphorus excess feeding.

Diet	Preprandial P/Crea	Postprandial P/Crea
CON	2.2 (1.0/2.8)	1.8 (1.3/2.2)
mCaP	5.8 (5.4/6.5) *	13.0 (9.0/17.0) ***
diCaP	6.0 (3.9/7.1) *	9.3 (7.6/10.9) ***
mNaP	6.3 (4.3/7.5) *	21.1 (16.4/25.9) ***
STTP	3.4 (3.1/4.2)	7.0 (4.1/9.9) **
mKP	6.7 (5.8/8.6) *	21.8 (17.1/26.5) ***
KpyrP	3.5 (3.2/4.3)	9.9 (6.7/13.1) ***
PM	4.3 (3.6/4.9)	1.5 (0.8/2.2)
CBM	5.8 (4.6/6.9) *	2.9 (1.5/4.2)

Median (IQR); * Significant difference to control group (* *p* < 0.05, ** *p* ≤ 0.01 and *** *p* ≤ 0.001).

## Data Availability

Data presented in this study are available on request from the corresponding author.

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
