# Peer review of "The Source Matters–Effects of High Phosphate Intake from Eight Different Sources in Dogs"

_animals, 2021, doi:10.3390/ani11123456_

Round 1
Reviewer 1 Report
M&M section lacks information on:
- ethical approval of the study
- the housing of the dogs. Urine samples were collected (L151) and faeces were but it is not stated how this was done. Was there possibility for coprophagia? How were urine samples collected?
- general information on the amount of food provided (L121-122 states that individual energy requirements were met but not that feed offered was adjusted during the trial), feeding rate, refusals, preparation of food before feeding, any processing of the food, …...
- the order in which the diets were tested. Was this conform the list in Table 1 (except Con diet)?
L80, suggested to replace “huge” with for example “significant” or “large”.
L81, delete “about”.
L82, replace “workload” with a better description of the processes performed by a nephron.
L85, The major regulator in the body ……
L94, where instead of were.
L97, leads instead of lead.
L120, how were these “individual requirements” determined?
L125, “on to”?
L126 and other line in the manuscript, add spaces between the value and unit.
L134, lard was added to the diet of some dogs.
L135, suggested to replace “basic diet” with “control diet” as used in Table 1, throughout the manuscript.
L145, how was it achieved to have particles <1 mm?
L152 were instead of was.
L184, there was apparently a feeding time which was extended. There was no mention of a feeding time in the M&M section.
L216, extent.
L231, there is not mention of the method used to measure creatinine in the M&M section.
Conclusion section (L339-346), these are not conclusions but is rather Discussion.
Title Table 2, Apparent digestibility of phosphorus and calcium in adult dogs fed diets containing various sources of phosphorus.
Table 2 and others, There are no values with two or three stars m(Table 2), no 2 stars in Table 4. So why mention them in the footnote? Explain abbreviations. What are the two values between brackets?
Table 4, these values are all highly similar and without SEM values, there is no way to see if the significance values are correct. Add SEM values to the table.
Author Response
Point by point reply
Dear reviewers, thank you very much for taking the time and helping to improve the manuscript. Please find the reply to your points below.
Reviewer 1:
M&M section lacks information on:
- ethical approval of the study
This information is given in line 354:
Institutional Review Board Statement: All procedures and protocols were conducted in accordance with the European guidelines of the Protection of Animals Act. The study was approved by the representative of the Chair of Animal Welfare of the Faculty of Veterinary Medicine of the Ludwig-Maximilians Universität München as well as the Government of Upper Bavaria (reference number AZ 55.2-1-54-2532.4-12-12).
Shall this information be added to the M&M section too?
- the housing of the dogs. Urine samples were collected (L151) and faeces were but it is not stated how this was done. Was there possibility for coprophagia? How were urine samples collected?
A: The passage was amended accordingly (changes in red/italics):
The dogs were pair-housed in climate controlled indoor pens with access to outdoor runs of about 50m2 in known groups of 4 to 7 animals for at least 6 hours per day.
During the 5 d balancing trial faeces were collected quantitatively whenever defecation was spotted to reduce the probability of coprophagia, even though this was unlikely in the selected dogs. Urine was sampled about 2 hours pre- and 3 to 4 hours postprandially using a special scoop.
- general information on the amount of food provided (L121-122 states that individual energy requirements were met but not that feed offered was adjusted during the trial), feeding rate, refusals, preparation of food before feeding, any processing of the food, …...
A: The passage was amended accordingly (changes in red/italics):
Individual energy requirements were determined over a period of 10 weeks prior to the start of the study by identifying the energy necessary for body weight maintenance.
Regular weighing with adaptation of energy supply in case of body weight loss by adding lard ensured minimal weight changes with a constant intake of the basic ration.
In chapter “results” added: Overall, the dogs maintained their body weight (< 4% variation) and BCS.
The diet was produced from thoroughly mixed cooked tripe, cooked rice and casein […]
Refusals were weighed and recorded.
Plus: in the results section the following sentence was added: In all other trials, the dogs consumed the apportioned amount of food in the normal feeding time (~15 minutes).
- the order in which the diets were tested. Was this conform the list in Table 1 (except Con diet)?
A: No, the order was differently and is now given:
As phosphorus sources inorganic phosphate salts (monocalciumphosphate (Ca(H2PO4)2), dicalciumphosphate (CaHPO4, synonym: calcium monohydrogen phosphate), monosodiumphosphate (NaH2PO4), sodiumtripolyphosphate (Na5P3O10; STTP), monopotassiumphosphate (KH2PO4), potassiumpyrophosphate (K4P2O7)) and organic phosphates (poultry meal (PM), cattle bone meal (CBM)) were used in the following order: CON, diCaP, mNaP, PM, STTP, mCaP, CBM, mKP, KpyrP.
L80, suggested to replace “huge” with for example “significant” or “large”.
A: Done
L81, delete “about”.
A: done
L82, replace “workload” with a better description of the processes performed by a nephron.
A: The processes performed by a nephron are manifold. The word workload describes – at least in my understanding – what was planned to express: an increased use to capacity, ‘volume of work’. Not only the filtered load of phosphorus and other substances increases but all parts of the physiological activity within a nephron. Please advise in case this is not acceptable.
L85, The major regulator in the body ……
A: Adopted
L94, where instead of were.
A: corrected
L97, leads instead of lead.
A: corrected
L120, how were these “individual requirements” determined?
A: Amended by adding the following sentence: Individual energy requirements were determined over a period of 10 weeks prior to the start of the study by identifying the energy necessary for body weight maintenance.
L125, “on to”?
A: Corrected
L126 and other line in the manuscript, add spaces between the value and unit.
A: A space was added between value and unit.
L134, lard was added to the diet of some dogs.
A: Adapted
L135, suggested to replace “basic diet” with “control diet” as used in Table 1, throughout the manuscript.
A: Thank you, amended where applicable
L145, how was it achieved to have particles <1 mm?
A: The sentence was amended: To exclude a possible influence of particle size of the phosphorus source on phosphorus digestibility, it was ensured through grinding and sieving that
L152 were instead of was.
A: corrected
L184, there was apparently a feeding time which was extended. There was no mention of a feeding time in the M&M section.
A: Thank you for the note. The beagles normally need less than 5 minutes to consume the complete daily ration. We allowed for an average of 15 minutes feeding time. The information was added to the manuscript as follows (changes red/italic): Extension of feeding time, which normally amounted to less than 15 minutes to allow complete intake, and the reoffering of food led to a feed intake sufficient to prevent relevant weight loss.
L216, extent.
A: corrected
L231, there is not mention of the method used to measure creatinine in the M&M section.
A: Thank you. The following sentence was now added: Urine creatinine was determined using the Jaffé method (MicroVue Creatinine Assay Kit, Quidel Corporation, reader: Sunrise Tecan).
Conclusion section (L339-346), these are not conclusions but is rather Discussion.
A: The section was reworded:
This study demonstrated that in contrast to the more natural organic phosphate sources, excessive supply with the tested inorganic phosphates significantly disrupted elements of phosphorus homeostasis in healthy dogs. Therefore, adverse health effects of ingesting such inorganic phosphates cannot be excluded. Therefore, the use of inorganic phosphate sources, particularly mNaP, mKP and KPyrP, in dog and cat foods must be considered potentially hazardous.
Title Table 2, Apparent digestibility of phosphorus and calcium in adult dogs fed diets containing various sources of phosphorus.
A: Thank you for the note. Adopted
Table 2 and others, There are no values with two or three stars m(Table 2), no 2 stars in Table 4. So why mention them in the footnote? Explain abbreviations. What are the two values between brackets?
A: The explanations for the asterix are now only given in the footnote when the level of significance exists in the data.
The explanation for the values before and within the brackets were added to table 2 and the statistics section in materials and methods (Data are given as mean with interquartil range (IQR) in brackets.)
Table 4, these values are all highly similar and without SEM values, there is no way to see if the significance values are correct. Add SEM values to the table.
A: The data are given as median (IQR) and therefore without SD or SEM which is only used with normally distributed data. The differences in mean values are indeed small, as expected in serum calcium, but significantly different probably due to limited variation.

Reviewer 2 Report
In this paper, authors described the harmfulness of inorganic phosphate in a pool of beagles as a consequence of a daily pet food intake. Such results are important in light of the physiological homeostasis that is highly triggered and possibly impaired in case of pathologic or parapathologic conditions. The topic is important and has a novelty in the point of the continuous growing awareness of the role of a tuned supplementation in pets, as nutrition has been widely demonstrated to be the first cause of pathology onset.
However, the number of animals proposed by the authors is too small to draw any consistent conclusion
- I believe that readability would really benefit if authors would create subsections in the materials and methods section
- how many animals were kept? Housing conditions need to be provided
- line 116: animals were recruited or involved, not available
- all chemical formula should have their numbers as subscripts
- references style should be revised according to the journal guidelines
Author Response
Point by point reply
Dear reviewers, thank you very much for taking the time and helping to improve the manuscript. Please find the reply to your points below.
Reviewer 2:
In this paper, authors described the harmfulness of inorganic phosphate in a pool of beagles as a consequence of a daily pet food intake. Such results are important in light of the physiological homeostasis that is highly triggered and possibly impaired in case of pathologic or parapathologic conditions. The topic is important and has a novelty in the point of the continuous growing awareness of the role of a tuned supplementation in pets, as nutrition has been widely demonstrated to be the first cause of pathology onset.
However, the number of animals proposed by the authors is too small to draw any consistent conclusion
A: The number of dogs used in this trial was requested (application for ethical approval) on a biometrical calculation based on the expected differences between control and feeding group regarding selected parameters. The number of animals is not allowed to be higher than strictly necessary to achieve statistically significant differences. The same number of dogs were used for the study published by Dobenecker et al. 2021 and the results were confirmed. The authors interpret the repeated results as basis to draw conclusions.
- I believe that readability would really benefit if authors would create subsections in the materials and methods section
A: amended, materials and methods was divided into 4 subsections
- how many animals were kept? Housing conditions need to be provided
A: Added
The dogs were pair-housed in climate controlled indoor pens with access to outdoor runs of about 50m2 in known groups of 4 to 7 animals for at least 6 hours per day.
- line 116: animals were recruited or involved, not available
A: Changed
- all chemical formula should have their numbers as subscripts
A: amended
- references style should be revised according to the journal guidelines
A: Amended

Round 2
Reviewer 2 Report
Could you please provide details about climate controlled indoor pens ?
Humidity, temperaure, dark/light cycle...
Author Response
The passage was amended as follows:
The dogs were pair-housed in climate controlled indoor pens including sufficient resting places with bedding material. Every day of the year they had access to outdoor runs of about 50 m2 equipped with kennels as well as trees or awning, in established groups of 4 to 7 animals for at least 6 hours per day (during 8 a.m. and 4 p.m.). Additionally, the dogs were walked on a leash and trained in regular intervals. During the digestibility trials, they were walked at least twice daily. The indoor pens had natural and artificial light for a minimum of 8 hours per day depending on the season. Humidity varied between 40 and ~70%. Fresh air was provided through a ventilation system throughout the year. The indoor temperature was kept above 16°C.